# Living restricted lives - Understanding the impact of isolation, social distancing and other restriction measures on older care home residents and their relatives in England during the COVID-19 pandemic: A qualitative study

**Sarah Sims**[1]ᵒ, **Amit Desai**[1]ᵒ, **Ruth Harris**[1]ᵒ, **Anne Marie Rafferty**[1]ᵒ, **Shereen Hussein**[2]ᵒ, **Richard Adams**[3]ᵒ, **Lindsay Rees**[4]ᵒ, **Sally Brearley**[1]ᵒ, **Joanne M. Fitzpatrick**[1]ᵒ *

1 Florence Nightingale Faculty of Nursing, Midwifery and Palliative Care, King's College London, London, United Kingdom, 2 Faculty of Public Health and Policy, London School of Hygiene and Tropical Medicine, London, United Kingdom, 3 Voyage Care, London, United Kingdom, 4 Quality Compliance Systems, Hayes, United Kingdom

ᵒ These authors contributed equally to this work.
* joanne.fitzpatrick@kcl.ac.uk

**Data Availability Statement:** This manuscript reports qualitative data from care home residents

## Abstract

### Background

During the COVID-19 pandemic, care homes for older adults in England implemented isolation, social distancing and other restriction measures to help protect residents from contracting the virus. Little was known about the physical and psychological impacts that these measures would have upon residents and their relatives.

### Aim

To explore the experiences of residents and their relatives of living restricted lives during the pandemic.

### Methods

This qualitative study was conducted as part of a 12-month, mixed-methods, phased design. From six purposively sampled care homes in England, 17 purposively sampled residents (all older adults) and 17 purposively sampled relatives participated in an individual, on-line interview, where they discussed their experiences of the restrictive measures implemented within their care home. Interviews were audio- and video-recorded with participants' permission and analysed using an inductive orientation to thematic analysis, with coding and theme development driven by the data content.

### Results

Participants' experiences of care home restrictions varied; their impact was influenced by the existing pattern of relationships that residents and their relatives maintained within and

and their families. Sharing the data publicly would present an ethical risk that could compromise participant confidentiality and privacy, with potentially identifying or sensitive information. All relevant data are presented within the manuscript; the manuscript contains anonymised quotes from a range of different resident and family participants and respects the terms of consent by individuals who were involved in the research. Further information can be obtained from the corresponding author. The interview schedules are included as Supporting information (S1) and (S2).

**Funding:** This project was funded by the National Institute for Health Research Health and Social Care Delivery Research Programme (Project number: 132541) COVID-19 Recovery and Learning Programme. The views and opinions expressed therein are those of the authors and do not necessarily reflect those of the National Institute for Health Research Health and Social Care Delivery Research Programme or the Department of Health and Social Care.

**Competing interests:** The authors have declared that no competing interests exist.

beyond the care home. It was further influenced by the fact that many residents and relatives were still learning how to manage their relationships in the new context of living in a care home. Social distancing measures made care homes feel less homely and denied residents, staff and relatives physical touch and other forms of non-verbal communication. Many residents expressed a broad sense of gratitude that was associated with safety and well-being beyond the pandemic. As older adults, they put the pandemic, and its associated restrictions, within the larger context of their lives.

## Conclusions

Learning from the COVID-19 pandemic is paramount for governments, societies, policy makers, care home providers, care homes and their staff, residents and their families and friends, and researchers. Our study makes an important contribution as one of the first to study the impact of implementing isolation, social distancing and other restrictive measures for care home residents and their relatives in England.

## Introduction

The SARS-CoV-2 virus, also known as COVID-19, was declared a global pandemic by the World Health Organisation on 11 March 2020 [1]. In the United Kingdom (UK), the first national lockdown during the pandemic was announced on 23 March 2020, with the country being ordered by the British Government to "stay at home" and "save lives" [2]. On 15 April 2020, an action plan for social care in England was introduced by the Government that adopted a four-pillar approach aiming to control the spread of infection within care settings: support the workforce; support independence; support people at the end of their lives; and support local authorities and providers of care [3]. Plans for the other three countries of the UK occurred around the same time by the devolved administrations of Wales, Scotland, and Northern Ireland.

Social care in the UK includes care for older adults living in care homes which may be residential or nursing care homes or providing both residential and nursing care. Care is delivered by care staff and nurses (providing fundamental personal and health care), with external support provided by practitioners employed by the National Health Service (NHS). The care home sector is diverse and complex, for example, in terms of ownership, provision, size, and funding arrangements. Social care tends to be funded by people paying for their own care, by local government, or by a mixture of the two. Many older adults living in UK care homes have complex health and social care needs [4, 5], with approximately 75% living with a cognitive impairment [6]). Nearly half a million people live in care homes across the UK [7]. Pre-vaccination, older adults living in care homes were identified as being at high risk of poor health outcomes and mortality if they contracted COVID-19 [8], though this risk has been lowered by the COVID-19 vaccination programme [9]. The Office for National Statistics for England and Wales reported that between the week ending 20 March 2020 and the week ending 21 January 2022, 16.7% (45,632) of all care home resident deaths were attributable to COVID-19 [10]. Yet early evidence indicated that care homes caring for older adults were overlooked in the initial planning of how to contain COVID-19 [11] and there were reports of care homes facing significant challenges [12, 13].These challenges included inadequate support to manage infection prevention and control effectively; decision-making at speed in a vacuum of

evidence-informed guidance to care safely for residents; problems sourcing and funding personal protective equipment (PPE); concerns about testing; and a lack of guidance related to the discharge of older adults from hospitals to care homes [14, 15]. The high mortality rates in early waves of the pandemic necessitated swift and often drastic measures to be implemented to help protect residents from contracting COVID-19. These measures, as per Government guidance, included isolation, social distancing, and restrictions on visitors entering care homes, as well as restrictions on residents leaving [16]. These measures were required to be implemented despite limited evidence to support their delivery [8]. Further, little was known at the time about the physical and psychological impacts that these measures would have upon care home residents and their relatives.

While research into the effects of the pandemic on care home residents and their relatives is emerging e.g., [17–22], an understanding of the physical and psychological impacts of the restrictions imposed upon care home residents was limited when this study was conducted [23]. Few studies were undertaken at the time the restrictions were in force because of the challenges in undertaking research at that time. The evidence base to support the delivery of social distancing and isolation for older adults in care homes was lacking. This accentuated the need for research to explore and understand the challenges experienced by care homes endeavouring to implement these measures in a person-centred way that did not make home feel like an institution of confinement. It was critical to capture the expert ways in which care homes were managing well to implement social distancing and isolation requirements and mitigating negative consequences. For residents, potential negative psychosocial and physical repercussions included loneliness, low mood, loss of cognitive function [24], loss of physical function [8] and for those living with dementia, a worsening of both cognitive and psychological symptoms [25]. Possible adverse consequences for family and friends included loss and grief [26].

In 2021, our mixed-methods research study was undertaken as part of the National Institute for Health Research, Health and Social Care Delivery Research programme[27], which aimed to explore and understand the real-life experiences of isolation, social distancing and restriction from the perspective of residents, relatives, care home staff and managers, providing a rare insight into the experiences and perspectives of one of the most vulnerable groups i.e. care home residents as they were living with the restrictions. This study identified novel strategies and solutions adopted by the care homes to manage the mandated restrictions. Evidence-informed guidance and resources to support care homes, staff, and residents and their families and friends were co-produced with resident and public representatives using the findings of the study and evidence review [27]. This paper reports the detailed findings from the resident and family interviews.

### Aim

The aim of this paper is to explore the experiences of older care home residents and their relatives of living restricted lives during the pandemic.

## Materials and methods

### Design

Addressing study design criteria [28], a mixed-methods, three-phased design was undertaken for the funded study. The full research protocol has been published previously [27]. These phases comprised: Phase 1). a rapid evidence review of measures used to prevent or control the transmission of COVID-19 and other infectious diseases in care homes for older adults [29]; Phase 2). in-depth case studies of six purposively sampled care homes in England, involving remote individual interviews with staff, residents and families, the collection of policies,

protocols and routinely collected care home data, and remote focus groups with senior health and care leaders, and Phase 3) the development of evidence-informed guidance and resources for care homes that included a short film created from excerpts from interviews with residents, relatives, and care home staff detailing which interventions and strategies for social distancing and isolation worked well and which did not work in specific situations and contexts. This paper focusses upon findings from residents and their relatives from Phase 2 of the study. It is reported using consolidated criteria for reporting qualitative research (COREQ) [28].

## Setting

Data collection reported in this paper was undertaken between February and December 2021. A case study approach [30] was used to examine how social distancing and isolation of residents were being implemented in the six care homes. The care homes were purposively sampled to maximise variability in terms of size, type of care home, geographical location, ownership, incidence of COVID-19, and Care Quality Commission (CQC) rating. The CQC is the independent regulator of health and adult social care in England. It monitors and inspects services to assess whether they are safe, effective, caring, responsive and well-led and rates them as outstanding or good or requires improvement or inadequate [31]. In this study the care homes were situated across England, and all were part of larger organisations (ranging from seven to 114 care homes per organisation and between 767 and 5875 beds per organisation). Four of the participating homes were part of privately run organisations, and two were part of voluntary/not for profit organisations. One care home had a 'Dual' CQC registration (i.e. providing nursing and residential care), three had a 'Nursing' registration, and two were registered as 'Residential, without nursing'. All the participating care homes had a CQC rating of either 'good' or 'outstanding'.

The process of recruiting care homes began with an initial meeting with representatives of interested provider organisations, who had been sent the study information via existing contacts and networks of the research team. Following this meeting, the provider representative nominated a care home that they considered met the criteria, and that would have the capacity to take part in the study. A further meeting took place between the manager of the nominated care home and the researchers to provide more detail about the home's involvement in the research. All meetings took place remotely because of COVID-19 pandemic restrictions. Each participating care home manager was asked to nominate a 'project champion' to be the point of contact within the home and to help facilitate the research. The project champion was a member of staff who knew residents and their relatives well, and who had capacity in their role to help with the recruitment and interview process. The project champion role was undertaken by different staff members in each care home and included the manager, deputy manager, well-being coordinator, and administrator. The project champions were briefed about the study and guided by research team members throughout the process.

## Participants

Each care home was asked to purposively recruit three residents and three relatives, ensuring a range of genders, ethnicities, and different health and care needs. Inclusion criteria were that residents must be over 65 years and have the capacity to consent (as assessed by the care home manager). Relatives did not need to have a resident family member participating in the research to be involved themselves. Residents who met the inclusion criteria and relatives were nominated by the care home manager and invited to participate by the project champion, using paper copies of the study information sheets and consent forms. Information sheets and consent forms were tailored to each participant group; resident and relative documents were

produced in an easy-read format, following guidance from the Dementia Engagement and Empowerment Project (DEEP) (https://www.dementiavoices.org.uk/) and feedback from members of the study Patient and Public Involvement (PPI) Group. The latter comprised ten mostly older adults, several of whom had extensive personal experience of health and care services, and several who were or had been (informal) carers. All participants were given the option to have a phone or video call with the researcher before giving written consent, to ask any questions or talk through the research process; however, none chose to do so. Interviewers and participants had no relationship before the interview. Interviewers were aware of the names of participants before their interview, but transcripts were anonymized for the analysis.

## Data collections methods

Immediately before the interview began, the researcher checked participants had provided written consent, reminded the participant that participation was voluntary and that the interview could be paused or stopped at any time, and gave the participant the opportunity to ask any questions.

All interviews were conducted remotely using Microsoft Teams or VMix for those that were being video-recorded, due to COVID-19 restrictions on researchers visiting care homes at the time of the study.

All resident interviews took place within the care home, using an iPad sent to the care home by the research team. Interviews were conducted in either the resident's room or another quiet and private place in the care home. Relatives were given the option of doing their interview in their own home using their own device or within the care home using the research iPad. Only one family member chose to be interviewed in their own home and all other relative interviews were conducted within the care home, complying with visitation requirements. All participants were asked if they would like the project champion to be present during their interview for support; five residents and five family members chose this.

All participants were recruited, and interviews conducted between February and December 2021. Interviews were conducted non-simultaneously, and interviews at one care home were generally completed before interviews at another began. This sequential approach to data collection meant that data were collected at different times for different care homes. Addressing the research team and reflexivity [28], the interviews were conducted by four members of the research team (SS, SP, AD, JF). All interviewers are established academic researchers with experience in qualitative interviewing and a background in health and/or social care research. Interviews were semi-structured, and schedules were informed by findings from the rapid review of the literature (Phase 1), the research team, and reviewed by the study PPI Group, and a care home manager and a resident from a non-participating care home. When interviewing residents and family members, we asked about their own specific experiences of social distancing and isolation. Additional prompts were added to the schedules following the initial interviews in the first care home and were agreed by the whole research team. The schedule of questions was shared with each participant at the point of recruitment to the study so that participants had time to consider their answers (S1 and S2 Files).

A demographic form was completed with each participant before the recording began. For resident participants, these were collected with permission from residents and the care manager (e.g., about their primary health needs, length of time living in the care home, age group, gender, ethnic group). For families participating in the study, demographic data included the nature of their relationship to residents, age group, gender, and ethnicity. Demographic data were stored separately from interview transcripts. Interviews lasted between 20 minutes and one hour. All participants gave their written permission for interviews to be audio recorded.

Following each interview, the researcher made field notes about the engagement of participants, any key points that had arisen, and whether there had been any technical issues, such as problems with Wi-Fi; these reflexive notes were discussed at regular team meetings.

## Data analysis

Addressing analysis and findings [28], audio recordings were transcribed verbatim by an approved transcribing company and were quality assured by a researcher. Time restrictions meant that member checking of transcripts by participants was not possible. Data were analysed using an inductive orientation to thematic analysis, with coding and theme development driven by the data content [32]. Analysis was conducted by three experienced qualitative health and/or social care researchers (SS, AD, JF), and a sample of the transcripts were also read and analysed by other members of the research team (RH, SH) to ensure consistency of approach and enhance trustworthiness. The researchers met regularly to discuss their analysis, and to further enhance transparency and data trustworthiness, the manuscript contains anonymised quotes from a range of different resident and family participants. Themes were confirmed from multiple data sources from residents, relatives, and care home staff.

## Ethical considerations

The study was granted ethical approval by Coventry and Warwick Research Ethics Committee on 6[th] January 2021 (Project ID 291728). Permission to access the care homes was obtained as per local procedures. Informed written consent was obtained for all participants, and all participants were informed that they were free to refuse to participate or withdraw from the study at any time. A protocol was developed to manage any disclosure of poor practice and any distress of participants during data collection. The study Data Monitoring and Ethics Committee helped oversee the safeguarding of the rights and well-being of participants. For example, during the interviews with residents, our protocol was that should a researcher have any concerns about a resident's ability to consent to participate, they would not proceed. A modification of Dewing's (2007) process consent framework also helped to guide this process (See Table 1) [33, 34].

**Table 1. Dewing's process consent framework modified [34].**

| Stage | What is involved? |
|---|---|
| Preparation and background | Gaining 'permission' from gatekeepers to access the person if they are in a location where this is necessary.<br>Finding out about the person's biography and indicators of well-being.<br>Establishing basis for consent.<br>Understanding the person's capacity or ability to consent and facilitating this.<br>Establishing usual ways of expressing consent and signs of well-being and ill-being. |
| Initial consent | Considering consent and assent.<br>Using a range of written and visual prompts/information to enhance understanding.<br>Maintaining extensive notes as evidence to account for methods/decision/ |
| Ongoing consent | Monitoring.<br>Revisiting consent on each research encounter with the person and responding accordingly. |
| Feedback and support | Providing feedback to participant, care staff/family carers following research encounter (while respecting confidentiality of person living with dementia).<br>Assisting the person's transition from research encounter back into care environment.<br>Identifying any concerns to take to lead investigator. |

## Results

Individual interviews were conducted with 17 residents and 17 relatives of residents across the six care home sites. A summary of participating residents' and relatives' characteristics is provided in Table 2.

We were only able to recruit two family members at Care Home Five and two residents who had capacity to consent at Care Home Six. Care home residents and relatives shared with us their experiences of the restrictions that affected their lives and relationships during the COVID-19 pandemic. Key themes from these interviews were safety, practices of sociality, stoicism, and gratitude and these themes are presented below. To protect the anonymity of participants, quotations have not been assigned to specific care homes.

### Safety

Safety was a key theme for residents and relatives and there was an overarching belief that the participating care homes had successfully maintained the safety of their residents throughout the pandemic. Residents trusted care home staff and appreciated the sense of protection that living in a care home had offered them during this time:

> *"I think had I been outside and had I lived in a normal house and stuff, I might have been a bit frightened. But I thought 'I'm so well protected here'."* (Resident 14)

Relatives also reported the belief that residents were 'cocooned and protected' from the ways that people outside the care home were experiencing the pandemic. As one relative described:

**Table 2. Summary of participating residents' and relatives' characteristics.**

|  | Residents (n = 17) | Relatives (n = 17) |
|---|---|---|
| **Ethnicity** | White (n = 17) | White (n = 16)<br>Asian/British Asian (n = 1) |
| **Gender identity (as stated by participant)** | Female (n = 10)<br>Male (n = 7) | Female (n = 13)<br>Male (n = 4) |
| **Age (years)** | 35–44 (n = 0)   75–84 (n = 4)<br>45–54 (n = 0)   85–94 (n = 6)<br>55–64 (n = 1)   95–104 (n = 2)<br>65–74 (n = 2)   Missing data (n = 2) | 35–44 (n = 1)   75–84 (n = 1)<br>45–54 (n = 2)   85–94 (n = 0)<br>55–64 (n = 7)   95–104 (n = 0)<br>65–74 (n = 4)   Missing data (n = 2) |
| **Years at the care home** | 1 or less (n = 2)   5 (n = 2)<br>2 (n = 6)   6 (n = 1)<br>3 (n = 4)   Missing data (n = 1)<br>4 (n = 1) | N/A |
| **Receiving nursing care** | Yes (n = 9)   Missing data (n = 3)<br>No (n = 5) | N/A |
| **Dementia diagnosis** | Yes (n = 2)   Missing data (n = 1)<br>No (n = 14) | N/A |
| **Tested positive for COVID-19** | Yes (n = 5)   Missing data (n = 1)<br>No (n = 11) | N/A |
| **Placed in isolation** | Yes (n = 10)<br>No (n = 7) | N/A |
| **Relationship to participating resident** | N/A | Son or daughter (n = 2)<br>Spouse or partner (n = 2)<br>Sibling (n = 1)<br>Niece (n = 1)<br>Daughter-in-law (n = 1)<br>Not related to a participating resident (n = 10) |
| **Relationship to non-participating resident** |  | Son or daughter (n = 9)<br>Daughter-in-law (n = 1) |

*". . .the pandemic is sort of secondary to her life which is great".* (Relative 8)

This sense of safety was enhanced by the visiting restrictions that care homes put in place during the pandemic, which ranged over time from a total ban on outsiders, to visiting with enhanced measures such as testing, social distancing, and mask-wearing. Many accepted that these visiting restrictions had helped to successfully protect and maintain the safety of residents and in some cases, keep the care home completely free of COVID-19. That was not to say that visiting restrictions did not have negative implications for residents and indeed, some talked at length about the effect they had upon their mental health and well-being:

*"The worst thing was not seeing my family, that was terrible, absolutely terrible"* (Resident 13)

Relatives also talked of the effect it had upon their own mental health and well-being, as well as their relationships with their loved ones living in the care home:

*". . . we've just felt helpless because we had to sit back and watch mum go through a really traumatic time. . .and it was just heart-breaking that we just couldn't get in there and just give her a big hug, and hold her hand, which is all that she actually wanted from us to be honest".* (Relative 3)

When visiting was completely banned, both residents and their relatives valued being able to communicate with one another virtually and highlighted the importance of this for their health and well-being. Several innovative and creative approaches were identified to enable contact and communication, and stories were shared of how many residents had adjusted to using smartphones and tablet devices to keep in touch with their relatives:

*"[Mum] has been Skyping with children, grandchildren and so on, which was something that we actually didn't do before but is something that has actually been very meaningful for her and even when restrictions are off, I think that's something that will be a legacy"* (Relative 5)

Others used videocalls less often or not at all but made conventional audio calls. Residents with personal access to advanced video calling technology provided by their relatives–and with the skills to operate it - appreciated the autonomy it afforded them, as they did not need to rely on staff to facilitate calls and could contact friends and family at times that suited them. This contrasted with those who did not own a mobile phone or could not operate one, as videocalls worked less well for those unable to operate mobile technology or without their own devices. When total bans were lifted and visiting with enhanced measures was implemented, residents and relatives talked of having to balance their desire to be close to and physically touch their relatives, whilst appreciating the importance of keeping them safe. It became clear that living in this balance provoked profound moral dilemmas for some relatives, as they made decisions about the risks of contact in a situation of great uncertainty. On occasion, some relatives did admit to having physical contact with their family member in ways that were not officially permitted by the care home, but these decisions had not been made lightly:

*"I do socially distance for quite a lot of the time but when I'm in the room and I'm with Mum I can give her a hug. I can give her a hug, she needs it."* (Relative 17).

The uncertainty experienced by relatives was linked to constantly changing information about COVID-19 and the inconsistent (as it appeared to them) guidance about what was safe

or not. Relatives were not always sure how or why decisions around care home restrictions were made and whether these decisions were based upon suggested government guidance or legal obligation. For residents, uncertainty was increased when they felt that care home staff had not explained the restriction measures to them as thoroughly or as respectfully as they would have liked. One resident, for example, described how he felt infantilised by how restrictions had been imposed and maintained by staff:

> *"I don't like being talked to as a child. . .Quite a lot of the matters were done in a military fashion. . .There could be a lot more handling of people and explaining rather than just saying 'you sit there, you sit there'"* (Resident 10)

Another safety measure implemented in many care homes during the pandemic was the physical isolation of residents who were suspected or confirmed as having COVID-19. Residents were also required to isolate upon admission to the care home and when returning from hospital, and in two of our participating care homes, all residents had been confined to their rooms at the start of the pandemic. Periods of isolation lasted for 10 or 14 days, depending on policy at the time. These resident isolation measures elicited negative reactions from those who remembered experiencing them. One resident said she had been required to isolate within her bedroom at the care home for two weeks after returning from hospital. She was isolated in a different room to her own and staff would visit her to deliver meals, though they would not stay to chat with her, which she attributed to their risk of being infected. This resident had no mobile phone, and no phone was brought to her, which meant that friends and family could not contact her during this time. She vividly described how she felt when her isolation period ended:

> *"All I know is when they opened that door I ran into the lounge [laughs] sounds ridiculous doesn't it, it did have a bad effect I must say, I thought God Almighty I never want to have to go through that again"* (Resident 14)

Whilst some family members were aware that their relative had probably been placed in isolation at some point, they were often unable to talk about those periods precisely; for some, they seemed to become part of the general raft of restrictions faced by residents. They also sometimes failed to distinguish between formal isolation and more informal, self-imposed isolation, such as choosing to take meals in one's room rather than eat communally. However, some family members felt that a period of formal isolation had negatively impacted their relative's physical, mental, and emotional well-being:

> *". . . I think that may have affected her. Because the first time I saw her after the first lockdown she had deteriorated, physically and mentally. But then again, I think she probably would have done anyway. But I think the longer periods spent in her room probably did have an impact."* (Relative 4)

This quote highlights the uncertainty of knowing what effects isolation and other restrictive measures had upon care home residents during the pandemic. This was particularly the case for those residents with advanced cognitive decline who could not fully express how they were feeling or why.

## Practices of sociality

Residents' reflections of how their everyday lives had changed in the care home during the COVID-19 pandemic - and of the impact these changes had upon them–varied depending

upon the way in which they had used or imagined the care home space prior to the pandemic and their attitudes towards socialising within it. For example, one resident had maintained an active social life outside the care home prior to the pandemic but did not socialise or have close friendships with other residents. As such, pandemic restrictions on leaving the care home and having external visitors changed this resident's life significantly and had a negative impact upon her mental health and wellbeing:

> "... my husband and I had a very active social life in as, well you know, I used to go home Saturdays and Sundays for the day, and on Wednesdays we usually used to go out for a meal and coupled with the fact that he was here every afternoon anyway. So, my life has changed enormously." (Resident 5)

Some relatives echoed this negative impact that restrictions in external socialising and activities had on residents. As one relative who had enjoyed accompanying her mother to the cinema told us:

> "... we had a cinema session once a week and we used to always see the same people at the cinema, we could chat about the films and have an ice cream with them and things. So she'll miss that. So, yeah, I think she'll have missed the socialising quite a lot". (Relative 12)

These examples highlight the experience of residents who viewed the care home as a base from which to conduct and pursue their social lives beyond its walls. The care home itself was not necessarily a source of valued relations for these individuals; they were particularly affected by restrictions on external visitors and pursuits. However, several residents described how restrictions on leaving the care home affected them very little because they seldom left their bedrooms anyway. Others said they had good friendships with their fellow residents and enjoyed socialising inside the care home with them. We spoke individually to two residents who regarded each other as friends, and both said that pandemic restrictions had not adversely affected their relationship. Indeed, one reported that their friendship had grown stronger because of the shared challenges they had faced during the pandemic.

Although internal socialising and activities were generally prohibited or restricted in the early phases of the pandemic, these were reintroduced later to mitigate some of the effects of the restrictions on external visitors and activities. These internal activities included doing arts and crafts; watching entertainers over Zoom; celebrations for national memorial days, anniversaries, cultural and religious days/events; cake-baking, card-making, and word games, as well as going on virtual trips. Relatives remarked on the positive impact of these activities and the crucial role of the wellbeing/ activities coordinator in acting upon signs of depression, loneliness, or social isolation in residents:

> "A specialist team...was brought in... there was two young women and they were really good with Mum, and the other residents, and that did make a difference, that really did, that helped, I can say that.".(Relative 13)

However, the ability to participate in these internal activities was impacted by residents' levels of health and mobility. For example, one wheelchair user said that her ability to participate in activities in the care home depended upon how many other wheelchair users wanted to do the same activity, as staff could not escort everyone. Once again, such stories show how the effects of pandemic-related restrictions are inextricably linked to a broader set of challenges experienced by older adults in the latter years of their lives.

## Stoicism and gratitude

The third theme arising from residents and their relatives was the stoicism they demonstrated during the pandemic and the gratitude they expressed towards care home staff. This sense of stoicism and resilience was demonstrative of residents' broader attitude towards the pandemic and the consequent changes it brought to their lives:

*". . . I have felt a bit down, but you've just got to accept what's happening and carry on"* (Resident 3)

Some relatives also demonstrated similar stoicism:

*""It's worse than the war, you know, worse than the war," she says, "tell you what, got through the war, we'll get through this."* (Relative 13)

Residents and their relatives recognised and were grateful that care home workers had performed above and beyond the scope of their usual roles during the pandemic. They praised staff for coping with exceptional busyness, for keeping residents and staff free from infection, and for ensuring that residents remained well-treated and happy. Residents also expressed a broader sense of gratitude that was associated with safety and well-being beyond the pandemic. For example, one resident described how she was thankful that she had lived a long life, which she owed to the care she received from her care workers:

*". . . I didn't think I'd live this long, nowhere near, you know. And, when I first come in here, I thought, well, that's it, you know, the end is nigh. But it's taking a long time coming."* (Resident 2)

We suggest that such a sentiment needs to be considered to understand the experience of residents living restricted lives during the COVID-19 pandemic. It reminds us that the perspective of those who are conscious of being in the final years of their life may have different understandings of 'restrictions' to others. This resident seemed to put the pandemic, and its associated restrictions, in the context of her unexpectedly long life and as such, the restrictions did not dominate her perspective. This point was made even more strongly by the experiences of another resident who was approaching the end of his life. He described how he and his wife lived in the same care home but had separate bedrooms due to their different care needs. Despite being in the same home, care staff had not allowed them to see one another while restrictions remained in place and instead, they communicated through video calls. His health deteriorated over the pandemic, and he was admitted to hospital. Concerned that he would die in hospital, he asked to return to the care home to be closer to his wife and the care home organised this, for which he was grateful. While the restrictions on not seeing his wife might seem excessively harsh, he did not think so and was instead simply grateful to still be alive and to be back in the care home closer to his wife. As he put it:

*"'I just thank God I'm able to have some contact with her".* (Resident 12)

## Discussion

This paper has presented the experiences of care home residents and their relatives of the social distancing, isolation, and other restriction measures implemented within care homes during

the COVID-19 pandemic. As we have highlighted, these experiences were influenced to a great extent by the existing pattern of relationships that residents and their relatives maintained within and beyond the care home. It was also influenced by the fact that many residents and their relatives were still learning how to manage their relationships in the relatively new context of living in a care home.

Residents and relatives alike who we interviewed valued the work of their care homes in keeping residents safe during the pandemic and generally understood and accepted the measures implemented to maintain their safety. However, social distancing and isolation measures and restrictions on having visits from families and friends had negative consequences for many residents and their networks.

## Social distancing measures

Social distancing measures included residents and staff being required to maintain a two-metre distance from each other [16]. We identified negative consequences of social distancing measures, which included confusion for some residents and limitations to their social interaction. Similarly, other studies have reported residents feeling lonely and isolated [35–37]. Some residents, notably those living with dementia, found it difficult to understand why they could not sit close to others. Some were confused about why they could not have a hug from staff though staff were permitted to assist with personal washing and dressing. This caused upset for residents and staff alike. This concurs with other research findings [38] who reported on the negative impact of restrictions for residents living with dementia and their relatives. Care homes were perceived by staff as a resident's home rather than an institution, consequently some staff questioned whether social distancing was appropriate and if abiding by social distancing measures was always in the best interests of residents. Likewise, other studies have reported on conflicts between implementing restrictions for residents and delivering best care, with attention to human rights for the older person [39, 40].

## Isolation measures

In line with Government guidance [16], initially residents were asked to stay in their rooms for 14 days, though this was reduced to 10 days from January 2022. During this isolation period, all care and meals were delivered to residents' private rooms. A key finding of our study was the reported negative consequences of isolation for some residents' physical, psychological, emotional, and cognitive well-being. Some residents found that the isolation measures implemented during the pandemic negatively affected their mental health, emphasising the desirability and value of socialising for some residents, as reported in other research [35]. This was especially true of those residents who were unable to communicate with the outside world through technology during periods of isolation. Similar findings associated with the psychological impact of quarantine have been identified elsewhere [24].

For some residents, isolation was believed to have had contributed to their physical decline (e.g., because of reduced physical exercise, and not eating and drinking as well when dining alone), and to a decline in mental health (e.g., experiencing disturbing hallucinations). Relatives likewise thought that being in isolation had negatively impacted on residents' physical and psychological health. A rapid review of the psychological impact of quarantine has reported that quarantine was often associated with a negative psychological effect and longer quarantine (more than 10 days) with poorer psychological outcomes [24].

For those residents who recalled being in isolation, it had been a challenging experience and isolating in every sense of the word. Residents who had recently moved into the care home were considered by staff to have had a particularly difficult experience of isolation regulations.

Transitioning to living in a care home was challenging enough without having to isolate alone, not having the opportunity to see the care home as it usually operated, to participate in activities, and to interact freely with and get to know staff and residents. These factors are potential facilitators for older adults transitioning well to living in a care home [41]. Unintended consequences of isolation measures included potential new residents being put off from moving in because of isolation requirements, reluctance sometimes for residents to attend hospital appointments to avoid having to isolate on their return to the care home, and care home decision making about potential new residents (e.g., accepting only older people who were physically and mentally capable of isolating themselves and not those who walk with purpose).

Measures to make isolation less difficult for residents included ensuring that they were entertained purposefully with regular socially distanced visits from staff and various therapeutic resources to occupy their time, whilst working with residents' individual beliefs and values. 'Activity boxes' were created. Some care homes gave isolating residents a radio, Echo, or Alexa so that they could listen to music; an iPad or tablet so that they could contact family and friends; a TV to keep them entertained; and exercise equipment such as stretch bands, to keep active. Adequate staffing levels and a skill mix that includes sufficient Wellbeing and Activities Coordinators are vital to an infrastructure that can support these measures. Family members praised the crucial role of these coordinators in identifying and acting on signs of low mood for residents in isolation. Engagement of residents in safe and meaningful activities [22] and additional close monitoring were key strategies for supportive isolation in other care home research studies [42, 43]. Creative approaches to activities for residents not isolating were also evidenced (e.g., small group indoor gardening, yoga, karaoke, bingo, and quizzes), mitigating in part relatives' concerns that residents might die due to lack of social contact and activity [19].

Purposeful activities alone to support residents isolating was considered inadequate. Staff sitting with residents regularly to help prevent loneliness and improve well-being was an important study finding. This illustrates the importance of human connectedness and resonates with reports of the six senses which are required for outstanding care, including the need for older adults to experience a sense of security, continuity, and belonging [44]. Maintaining good communication with residents throughout their period of isolation was important, with staff comforting residents, explaining the reasons why they needed to isolate and encouraging them to persevere. Most residents shared that the measures and restrictions introduced at the care home were explained well to them by staff although some shared that staff were not always willing to discuss the measures as fully as they would have liked. Some residents shared that it was likely that staff themselves did not understand the reasons for measures and restrictions, a finding of our interviews with care home staff. Giving as much information as possible to those who are in isolation, their families and friends was a theme in the published literature [24]. A pre-COVID-19 study found that best care was evident in care homes where residents, families and staff worked together, and where there was a sense of community [45].

As well as measures and restrictions within the care home, residents had to comply with restrictions for leaving the care home (e.g., to visit their own home and other places with staff or family and friends), and for using facilities in the wider care home community (e.g., the gym and cinema). Factors known to enable older adults to settle into care home living include being able to continue valued social relationships and establish new relationships [41]. These restrictions and changes to social interaction had negative consequences for residents' mental health, with staff reporting that residents became withdrawn and introverted, and for residents' physical health (e.g., contributing to deconditioning because of a lack of exercise). Other research has reported similar findings, reporting that disruptions to residents' usual care routines and social interactions had negative consequences for their physical and emotional well-being [46–48].

Residents told us that their mental health had deteriorated because they could not see their friends and family in person. Similar anxieties and stresses were also experienced by family members during periods of visitor restriction. These findings are reflected in the wider literature, where relatives of older adults living in nursing homes in Sweden shared that residents were "lonelier and sadder" and restricted in their movements [49], and relatives not being able to continue to provide aspects of care [19, 35]. For relatives, they wanted more information about residents' everyday life without having to ask for it, and there were worries about resident well-being. The relatives we interviewed accepted the need for restrictions to keep their loved ones safe but were not always sure how or why decisions about guidance and implementation of these restrictions were made. Many were fearful that they might not get to see their family member in person again and were anguished at no longer being involved in the resident's daily life or in the life of the care home community. There were additional feelings of powerlessness for relatives whose loved ones had moved into care homes during the pandemic and could not be fully involved during this significant life transition.

## Enabling connections and communication

During periods of visitor restriction, residents and their relatives appreciated the support they received in communicating virtually and highlighted the importance of remote communication for their mental health and wellbeing. Dissatisfaction has been reported where remote communications were described as infrequent, poorly organised, not private, not functioning, or unavailable [50]. Governments and care home providers have been called to act with "moral urgency" to prioritise and address the technological capabilities of care homes [51], p547.

Implementing isolation, social distancing and other restrictive measures present moral dilemmas for care home staff, due to the potential tensions between risks and benefits, and how to balance facilitating person-centred care for residents (particularly those with cognitive impairment) with a need to implement measures to help prevent and control COVID-19 outbreaks [52, 53]. Developing staff knowledge and skills to support the implementation of isolation and other restrictive measures to include ethical principles for decision-making warrants further attention. For example, a Dementia Isolation Toolkit was developed, providing ethical guidance on how to isolate people living with dementia safely and with attention to their personhood [42], whilst others have advocated an approach to care that is "principle-based, coupled with case-by-case application of the principles for individual residents." [54], p4. This highlights the need for senior leadership, care home managers and staff to understand key sources of law and their application to resident care, including the Mental Capacity Act (2005), the Human Rights Act (1998), the Equality Act (2010), and the Coronavirus Act (2020). The consequences, challenges, and solutions to implementing restrictive measures for care home residents illuminate the knowledge, skills and values required to be able to care well for residents and emphasises the need for further investment in the development and recognition of this workforce.

Finally, this study highlighted the sense of stoicism, resilience and gratitude that was demonstrative for many care home residents of a broader attitude towards the pandemic and the consequent changes it brought to their lives. Many of these older adults were aware that they were in the twilight of their lives and they put the pandemic, and its associated restrictions, within the larger context of their lives, offering understandings of 'restrictions' with meanings different to those experienced by younger, able-bodied people. We suggest that such a sentiment needs to be considered further to fully understand the experience of care home residents and their relatives of living restricted lives during the COVID-19 pandemic.

## Strengths and limitations

A key strength of this study was that several care home residents and relatives were successfully recruited and interviewed, despite the difficulties faced by the care home sector during the COVID-19 pandemic. Interviews were conducted between February and December 2021 and the sequential approach to data collection generated rich data about the experiences of social distancing, isolation, and other restriction measures during different stages of the pandemic. The data presented in this paper also has the following limitations. First, all participating care homes had a CQC rating of either 'good' or 'outstanding' and we were unable to recruit any care homes rated as 'requires improvement' or 'inadequate', which may limit the generalisability of the findings. Fieldwork was conducted remotely due to restrictions on visitors to care homes. Researchers were therefore not able to observe the different settings (e.g. resident rooms, virtual communication methods) or interact in a range of ways with the different participant groups. Secondly, residents who met the inclusion criteria and their relatives were nominated by the care home manager and invited to participate by the project champion. This was the only way to recruit residents and relatives as restrictions meant researchers were unable to enter the care homes, however, it is possible that this may have influenced the residents that were nominated to participate in the study. Residents varied in terms of gender, age, length of time living in the care home, health and care needs, their experience of isolation, and experience of testing positive for COVID-19. For 13 of the family participants, the resident was living with dementia or other cognitive impairment, though for 14 of the resident participants it was recorded that there was no formal diagnosis of dementia. Overall, the participant groups generated a rich account of their experiences and perceptions about the research question. Furthermore, despite an explicit effort to recruit residents and family members from a range of ethnic backgrounds, all residents and all but one of the family members we spoke to were White. We suggest that future research evaluates social distancing, isolation, and other measures and restrictions deployed in care homes for older adults, taking account of the effects of such measures on residents and their families. This will help prepare for any future pandemic. We also suggest that further research is done to develop and evaluate remote social interaction for residents living with dementia and their families and friends.

## Conclusions

A substantial number of older adults living in care homes died during the COVID-19 pandemic. The findings of this study need to be considered to fully understand the experience of care home residents and their relatives of living restricted lives during the COVID-19 pandemic. Care homes implemented innovative approaches to social distancing and isolation with varying degrees of success. Learning from these experiences is paramount so that support can be put in place to ensure care homes can recover but also do not have to endure what they did during the COVID-19 pandemic for any future pandemics. This learning can support care home providers with evidence-informed guidance that sets out what and how social distancing, isolation, and other restrictive measures could be operationalised to achieve person-centred and relationship-centred care for residents, many of whom are approaching the final years of their lives and living with dementia, and their families and friends. Our study can make an important contribution to this guidance as one of the first to study the impact of implementing these restrictive measures for care home residents and their relatives in England. For health and care policy makers, there are important implications about how guidelines are developed and disseminated to the care home sector, reinforcing the need for social care to be integral to health and social care systems, to ensure that actions taken during national emergencies fully account for the impact on all services and all settings.

Future research can include evaluation of evidence-informed guidance, developing a trauma-informed approach to caring for the care home sector, and investigating digital technologies to help residents with different health and care needs stay connected with families and friends.

## Supporting information

**S1 File. Interview schedule for residents.**
(DOCX)

**S2 File. Interview schedule for family members.**
(DOCX)

## Acknowledgments

We would like to thank the residents and relatives who shared their stories so openly and honestly as part of this study, helping us to learn from their experience of COVID-19. We also offer our sincere thanks to the care home providers, managers and staff who took on the role of a project champion, and all the staff who supported our research. We acknowledge with thanks the contributions of Sinead Palmer for her research assistance with ethics, recruitment, and interviews with residents, families, and care home staff, and to Andy Richards, Channel Director, and the KMTV team for video-recording interviews with residents, families, and care home staff. Finally, many thanks to the members of the study Steering Committee, Data Monitoring and Ethics Committee, PPI Group, and the co-design workshop participants for their insightful contributions.

## Author Contributions

**Conceptualization:** Ruth Harris, Anne Marie Rafferty, Shereen Hussein, Richard Adams, Lindsay Rees, Sally Brearley, Joanne M. Fitzpatrick.

**Data curation:** Sarah Sims, Amit Desai, Ruth Harris, Anne Marie Rafferty, Shereen Hussein, Richard Adams, Lindsay Rees, Sally Brearley, Joanne M. Fitzpatrick.

**Formal analysis:** Sarah Sims, Amit Desai, Ruth Harris, Anne Marie Rafferty, Shereen Hussein, Richard Adams, Lindsay Rees, Sally Brearley, Joanne M. Fitzpatrick.

**Funding acquisition:** Ruth Harris, Anne Marie Rafferty, Shereen Hussein, Richard Adams, Lindsay Rees, Sally Brearley, Joanne M. Fitzpatrick.

**Investigation:** Sarah Sims, Amit Desai, Ruth Harris, Anne Marie Rafferty, Shereen Hussein, Richard Adams, Lindsay Rees, Sally Brearley, Joanne M. Fitzpatrick.

**Methodology:** Amit Desai, Ruth Harris, Anne Marie Rafferty, Shereen Hussein, Richard Adams, Lindsay Rees, Sally Brearley, Joanne M. Fitzpatrick.

**Project administration:** Sarah Sims, Amit Desai, Joanne M. Fitzpatrick.

**Resources:** Shereen Hussein, Joanne M. Fitzpatrick.

**Supervision:** Ruth Harris, Anne Marie Rafferty, Shereen Hussein, Richard Adams, Joanne M. Fitzpatrick.

**Validation:** Sarah Sims, Amit Desai, Ruth Harris, Anne Marie Rafferty, Shereen Hussein, Lindsay Rees, Sally Brearley, Joanne M. Fitzpatrick.

**Visualization:** Sarah Sims, Amit Desai, Ruth Harris, Anne Marie Rafferty, Shereen Hussein, Richard Adams, Lindsay Rees, Sally Brearley.

**Writing – original draft:** Sarah Sims, Amit Desai, Ruth Harris, Anne Marie Rafferty, Shereen Hussein, Joanne M. Fitzpatrick.

**Writing – review & editing:** Sarah Sims, Amit Desai, Ruth Harris, Anne Marie Rafferty, Shereen Hussein, Richard Adams, Lindsay Rees, Sally Brearley, Joanne M. Fitzpatrick.

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
