## [Decision Letter · Decision Letter 0]

24 Jul 2023

PONE-D-23-08897Living restricted lives: understanding the impact of isolation and restriction on care home residents and their relatives in England during the COVID-19 pandemicPLOS ONE

Dear Dr. Fitzpatrick,

Thank you for submitting your manuscript to PLOS ONE. After careful consideration, we feel that it has merit but does not fully meet PLOS ONE’s publication criteria as it currently stands. Therefore, we invite you to submit a revised version of the manuscript that addresses the points raised during the review process.

We look forward to receiving your revised manuscript.

Kind regards,

Francesco De Micco, M.D., Ph.D.

Academic Editor

PLOS ONE

Journal Requirements:

Reviewers' comments:

Reviewer's Responses to Questions

**Comments to the Author**

1. Is the manuscript technically sound, and do the data support the conclusions?

Reviewer #1: Yes

Reviewer #2: Yes

2. Has the statistical analysis been performed appropriately and rigorously? 

Reviewer #1: No

Reviewer #2: N/A

3. Have the authors made all data underlying the findings in their manuscript fully available?

Reviewer #1: Yes

Reviewer #2: Yes

4. Is the manuscript presented in an intelligible fashion and written in standard English?

Reviewer #1: Yes

Reviewer #2: Yes

5. Review Comments to the Author

Reviewer #1: This paper focuses upon the direct experiences of residents and their relatives of living restricted lives during the pandemic. I think that the focus of this study could be interesting. Congratulations for your work and effort. However, I would like you to pay attention to some indications for improvement its importance.

Reviewer #2: Thank you for the opportunity to review this paper which reports the experiences of care home residents and relatives during periods of social restriction during COVID-19. It is an interesting read that not only highlights the oft reported concerns of relatives but also the resilience and acceptance of residents.

I have a few small queries.

Sampling

How were care home contacted. You say they were nominated by a provider so I wonder did you have variety of provider?

Recruitment I like the idea of a project champion especially in care home research where recruitment can be very challenging. I wonder how capacity to give consent was assessed and were their residents who expressed an interest but were found not to have capacity.

There should be mention of limitations of sampling which was through care home gatekeepers, Even though I fully accept this was the most practical way during the pandemic

Results

The results are well presented and provide new insights into the experiences of residents in UK care homes

Discussion.

A little more nuanced consideration of the results would improve the discussion. For example page 15 line 320 states resident and relatives appreciates staff efforts for purposeful entertainment yet the results identifies that poor mobility could hinder residents involvement. I wonder were there any themes of inequity coming through the data. I was also struck that some residents could independently use video or phone technology but others relied on staff.

The published protocol paper for this study mentioned the toolkit it might be helpful if the final part of the discussion makes readers of this paper aware of next steps.

Conclusion

Please review the conclusion the first sentence ifs not clear ‘the loss of older people’ is this we ‘lost older people’ relating to statistics on deaths, or the older people experienced this type of loss due to ..’

6. PLOS authors have the option to publish the peer review history of their article (what does this mean?). If published, this will include your full peer review and any attached files.

Reviewer #1: No

Reviewer #2: No

---

## [Author Response · Author response to Decision Letter 0]

15 Sep 2023

Thank you for your helpful reviewer comments. We have responded to each comment in the uploaded 'response to reviewers' table.

---

## [Decision Letter · Decision Letter 1]

31 May 2024

PONE-D-23-08897R1Living restricted lives - Understanding the impact of isolation, social distancing and other restriction measures on older care home residents and their relatives in England during the COVID-19 pandemic: a qualitative study.PLOS ONE

Dear Dr. Fitzpatrick,

Thank you for submitting your manuscript to PLOS ONE. After careful consideration, we feel that it has merit but does not fully meet PLOS ONE’s publication criteria as it currently stands. Therefore, we invite you to submit a revised version of the manuscript that addresses the points raised during the review process.

Please revise your manuscript in line with reviewer requirements for greater depth and detail across the methods and discussion sections. Please submit your revised manuscript by Jul 15 2024 11:59PM. If you will need more time than this to complete your revisions, please reply to this message or contact the journal office at plosone@plos.org. Please include the following items when submitting your revised manuscript:A rebuttal letter that responds to each point raised by the academic editor and reviewer(s). You should upload this letter as a separate file labeled 'Response to Reviewers'.A marked-up copy of your manuscript that highlights changes made to the original version. You should upload this as a separate file labeled 'Revised Manuscript with Track Changes'.An unmarked version of your revised paper without tracked changes. You should upload this as a separate file labeled 'Manuscript'.

We look forward to receiving your revised manuscript.

Kind regards,

Rosemary Frey

Academic Editor

PLOS ONE

Reviewers' comments:

Reviewer's Responses to Questions

**Comments to the Author**

1. If the authors have adequately addressed your comments raised in a previous round of review and you feel that this manuscript is now acceptable for publication, you may indicate that here to bypass the “Comments to the Author” section, enter your conflict of interest statement in the “Confidential to Editor” section, and submit your "Accept" recommendation.

Reviewer #3: (No Response)

Reviewer #4: (No Response)

2. Is the manuscript technically sound, and do the data support the conclusions?

Reviewer #3: Yes

Reviewer #4: Partly

3. Has the statistical analysis been performed appropriately and rigorously? 

Reviewer #3: N/A

Reviewer #4: No

4. Have the authors made all data underlying the findings in their manuscript fully available?

Reviewer #3: No

Reviewer #4: No

5. Is the manuscript presented in an intelligible fashion and written in standard English?

Reviewer #3: Yes

Reviewer #4: Yes

6. Review Comments to the Author

Reviewer #3: This manuscript reports on interviews with 17 residents and 17 relatives from six different care homes in England about their experiences living under restrictive measures in care homes during the COVID-19 pandemic.

The authors describe how the impact of restrictions varied among participants, heavily influenced by the nature of pre-existing relationships between residents, their relatives, and the care home community. Introducing social distancing measures contributed to a less homely atmosphere within the care homes, significantly affected aspects of care like physical touch and non-verbal communication between residents, staff, and relatives. Nonetheless, many residents acknowledged a sense of gratitude, appreciating the safety and well-being provided by the measures when placed in the broader context of their lives. Authors conclude that understanding the experiences of care home residents and their relatives during the pandemic is crucial for comprehending the full impact of living under such restrictions. It highlights the need for future considerations on the physical and psychological effects of pandemic-related measures in care homes.

Firstly, I would like to commend you on the significant effort put into this study on the experiences of care home residents and their relatives living under restrictive measures during the COVID-19 pandemic. The qualitative insights provided by interviews with 17 residents and 17 relatives across six care homes in England has the potential to enrich our understanding of the pandemic's impact on this vulnerable population.

To strengthen your manuscript further, I have several suggestions aimed at enhancing its clarity, depth, and relevance to the broader research community, centering around literature engagement, design and conceptual grounding.

The study is motivated by "an understanding of the physical and psychological impacts of the restrictions imposed upon care home residents" being limited. Many studies have explored the physical and psychological impacts of pandemic restrictions. A more detailed discussion of how your study builds upon, diverges from, or fills gaps in this literature would highlight its scientific value and theoretical implications more clearly. Here, the cited literature seems to be a bit thin, as there are several studies in the record now that have explored this and related issues (such as how technology for social interaction was put to use during the pandemic). I would like to see your stated motivation for the paper more clearly engaging with this research record.

I agree that such experiences are worthwhile documenting, but what is the scientific value of describing them (besides bringing them into the light)? Do the authors see any particular implications from this knowledge? Are there conceptual or theoretically interesting consequences from this knowledge? In the revised version, the authors give a high level description of how the findings can inform discussions, guideline development etc. at the end. This strikes me as a missed opportunity, and can be used to better motivate the article (and the problem space it is addressing) up front. In other words, the article needs to better answer the "so what" question.

One of the former reviewers asked for a better grounding in terms of research design, which I feel is not fully addressed yet. The revised version contains a section on design, where it is clear that this study emerges from Phase 2. However, that phase also mentioned a lot of other materials, in addition to the interviews here described. What is the relationship between this material, and the other aspects of the study (cursory mention is made of the "toolkit", but the reader does not really get a good grasp of what this is. There is additional qualitative materials that seems relevant from Phase 2, which is not engaged with here. Further clarification on the research design, particularly the relationship between the interviews conducted and other materials from Phase 2 of your study, would enrich the reader's understanding. Describing the toolkit and additional qualitative materials mentioned would offer a more comprehensive view of your research approach.

Moreover, what is meant by "direct experiences" by the authors? This appears to be a potentially theoretically loaded concept, but this use is not situated within a broader stream of literature. I assume the authors aim to describe some sort of first-person experiences about the pandemic restrictions - but they do not provide any kind of theoretical or conceptual grounding of these experiences (e.g., narrative research, phenomenology, etc.) - I don't have a particular conceptual axe to grind here, but there is definitely literature that could be engaged with here to make the paper more interesting. Without a grounding of this kind, it is a bit difficult to assess the kind of experiential claims being reported, as the qualitative material - e.g. use of quotations- is are fairly modest (probably due to restrictions on space).

I am wondering to what extent the themes identified by the authors reflect the interview guide, which is not uncommon in interview-based studies of this kind. How did the interview guide look? What is the relationship between the themes, and the structure of the interview guide? How many codes did the authors encounter, what where the coding groups you ended up with (are the ones presented here exhaustive, etc.)? A table would be helpful, alongside more information about what the content of the interviews revolved around. The best qualitative research - based on thick descriptions - gets us beyond the themes that can be predicted in advance - based on the interview guide for instance. More clearly showing the relationship between these themes and your codes might add further depth.

The conclusion could be stronger - 'these findings need to be considered to fully understand the experience of care home residents and their relatives of living restricted lives during the COVID-19 pandemic'. Again, so what? What are the general insights of interest here, for caring sciences or health services research? The revised version is more explicit about this issue, but there is still potential to draw this out more clearly. The toolkit that was developed as part of the project, for instance, might hold some relevance here. The conclusion could be strengthened by articulating the general insights of your findings more explicitly and discussing their relevance to caring sciences or health services research. Mentioning the toolkit developed as part of the project and its potential applications could provide a compelling case.

Finally, considering the specific insights and implications of your study, help clarify the intended audience for this contribution.

Reviewer #4: Manuscript – PONE-D-23-08897R1 – Reviewer Comments

General Overview

Thank you for the opportunity to review this manuscript entitled, Living restricted lives - Understanding the impact of isolation, social distancing and other restriction measures on older care home residents and their relatives in England during the COVID-19 pandemic: a qualitative study.

This qualitative study, a subcomponent of a mixed methods study, sought explore the experiences of residents of care homes and their families when living with restricted lives during the COVID-19 pandemic due to isolation and social distancing measures. Overall, the topic is important, and the information has the potential to provide a valuable contribution to the literature.

Specific Comments

Abstract

Please indicate in methods if all resident participants were older adults.

Introduction

Pg. 3, line 48: Please briefly define what is meant by “social care”, this is not a universally understood model of care. There also needs to be a description of how this care model is funded.

It would also be helpful for the authors to clarify who the residents living in social care are. For example, do the majority have dementia? How extensive are their care needs?

Pg. 3, line 55: Please clarify context. Are older adults at risk of poor outcomes if they are vaccinated against COVID-19. If authors are referring to pre-vaccine times, they need to clarify.

Pg. 3, line 56: Please provide an exact date for the statistic of 16.7%.

Suggest authors briefly discuss the fact that the swift and high mortality rates associated with COVID-19 in the early waves necessitated drastic measures (e.g., isolation) despite little research existing for their potential collateral impacts on social/mental well-being. Context is very important for understanding this research study.

Pg. 5, line 89: Please clarify what the film was for.

Materials and Methods

Please provide exact dates when this study was executed at the beginning of the methods section.

Who determined whether residents had the capacity to consent? Please clarify.

Did residents require support to use the iPad or was this part of the inclusion criteria that they be able to manage the device alone.

How were questions for qualitative interviews developed? Were they informed by a theoretical framework or model? Please include full interview guide as an appendix.

Methods section would be strengthened by discussing the authors positionality with regards to the research and discussing how they maintained reflexivity throughout the study processes.

More information is required regarding the case study approach utilized by the authors. What are the methodological underpinnings of this approach? Please provide more information including appropriate references.

More information required about how study rigor was ensured required. Using quotes from different participants is standard within qualitative research, this in and of itself doesn’t necessarily enhance the trustworthiness of the data.

Results

Did the authors collect information on other genders or only male/female as options?

Please provide a brief statement at the beginning of the results section stating what the main themes are prior to discussing them.

Quotes should be separated by text not piggy backed onto one another within the results.

Pg. 12 & 13: large section of description without quotes to support these findings. More exemplars from interviews required to make these claims.

Pg. 16: more exemplar quotes require to support perspectives made by authors.

Discussion

This section could be strengthened with inclusion of more literature particularly in section entitled isolation measures. There is a growing body of literature that speaks to the impacts of social isolation but also resilience among this population.

Strength and Limitations

Pg. 24, line 515: This is not a strength, since interviews were cross-sectional and member checking was not done. Therefore the participants were not retained over time.

Conclusion

The authors should not discuss the toolkit in the conclusion as it is not discussed anywhere else in the manuscript.

Please review manuscript for grammatical areas (multiple periods) and repetitive language e.g., last paragraph use “our findings” three times in close succession.

7. PLOS authors have the option to publish the peer review history of their article (what does this mean?). If published, this will include your full peer review and any attached files.

Reviewer #3: No

Reviewer #4: No

---

## [Author Response · Author response to Decision Letter 1]

5 Aug 2024

Please see response to reviewers table attached.

---

## [Decision Letter · Decision Letter 2]

9 Oct 2024

Living restricted lives - Understanding the impact of isolation, social distancing and other restriction measures on older care home residents and their relatives in England during the COVID-19 pandemic: a qualitative study.

PONE-D-23-08897R2

Dear Dr. Fitzpatrick,

We’re pleased to inform you that your manuscript has been judged scientifically suitable for publication and will be formally accepted for publication once it meets all outstanding technical requirements.

Kind regards,

Rosemary Frey

Academic Editor

PLOS ONE

Additional Editor Comments (optional):

Reviewers' comments:

Reviewer's Responses to Questions

**Comments to the Author**

1. If the authors have adequately addressed your comments raised in a previous round of review and you feel that this manuscript is now acceptable for publication, you may indicate that here to bypass the “Comments to the Author” section, enter your conflict of interest statement in the “Confidential to Editor” section, and submit your "Accept" recommendation.

Reviewer #4: All comments have been addressed

2. Is the manuscript technically sound, and do the data support the conclusions?

Reviewer #4: Yes

3. Has the statistical analysis been performed appropriately and rigorously? 

Reviewer #4: Yes

4. Have the authors made all data underlying the findings in their manuscript fully available?

Reviewer #4: Yes

5. Is the manuscript presented in an intelligible fashion and written in standard English?

Reviewer #4: Yes

6. Review Comments to the Author

Reviewer #4: (No Response)

7. PLOS authors have the option to publish the peer review history of their article (what does this mean?). If published, this will include your full peer review and any attached files.

Reviewer #4: No

---

## [Editor Report · Acceptance letter]

21 Nov 2024

PONE-D-23-08897R2 

PLOS ONE

Dear Dr. Fitzpatrick, 

I'm pleased to inform you that your manuscript has been deemed suitable for publication in PLOS ONE. Congratulations! Your manuscript is now being handed over to our production team.

Kind regards, 

on behalf of

Dr. Rosemary Frey 

Academic Editor

PLOS ONE